# Effect of the COVID-19 Pandemic on Paediatric Check-Ups and Vaccinations in Germany

**DOI:** 10.3390/vaccines11040720

**Published:** 2023-03-23

**Authors:** Rudolf Lindinger, Hartmut Richter, Thorsten Christian Reuter, Tim Fischer

**Affiliations:** 1Medical Affairs Department, MSD Sharp & Dohme GmbH, 81673 Munich, Germany; 2IQVIA Commercial GmbH & Co. OHG, 60549 Frankfurt, Germany

**Keywords:** paediatric check-ups, paediatric vaccinations, vaccines, COVID-19 pandemic, SARS-CoV-2, IQVIA^TM^ disease analyser, Germany

## Abstract

Paediatric check-ups and vaccinations are provided and free of charge in Germany. Despite being hitherto generally well-received and adhered to, it is possible that the lockdown implemented due to the COVID-19 pandemic resulted in delays or even cancellations of critical paediatric visits with healthcare providers. This study attempts to quantify the rate and time to follow-up for check-ups in Germany using the retrospective IQVIA^TM^ Disease Analyzer database. Additionally, timely administration of 4 vaccines (Hexavalent, pneumococcal, MMR-V, Rotavirus) was analysed to examine the impact of pandemic restrictions on vaccine uptake. The timeframes which were compared to determine the effects of COVID-19 were June 2018–December 2019 and March 2020–September 2021. The follow-up rates for paediatric check-ups were consistently lower in the COVID-19 phase, but generally ~90%. Follow-up rates for the vaccinations were distinctly higher during COVID-19. The time between events was almost unchanged for check-ups during the pandemic. For check-ups, age at initial event differed by less than a week between the phases. For vaccinations, the age differences were slightly higher, but exceeded one week in only two cases. The results show that the COVID-19 pandemic had little effect on paediatric check-ups and vaccinations in Germany.

## 1. Introduction

The German healthcare system ensures a high standard of care for its population, including for children. All prescriptions are reimbursed, even if normally over the counter (OTC), and free paediatric check-ups and vaccinations are provided. At each check-up, the child is examined for possible serious illnesses and their development is checked. Depending on the age of the child, the focus is placed on different areas: for example, mobility and dexterity, speech and comprehension, social behaviour, etc. In addition, depending on the age of the child, there are additional topics and counselling focuses, such as vaccination protection, prevention of Sudden Infant Death Syndrome (SIDS), accident prevention, dental health, and nutritional issues (Table 1). Vaccinations have a direct and proven beneficial effect, not only preventing the diseases they target but also possessing possible secondary benefits [1] (e.g., reducing the occurrence of male infertility following mumps [2]). Table 2 shows a summary of vaccinations recommended by the STIKO [3] for children in the first 18 months of a child’s life. Vaccinations are recommended to take place at as early an age as the vaccine is indicated for, but can be given at any age up to 18 years and still be reimbursed; thereafter, the reimbursement may be restricted.

Paediatric check-ups are, in some cases, timed to coincide with recommended vaccinations, and, in contrast to these, their reimbursement is regulated more strictly in Germany. There is a recommended age (in weeks) at which each check-up should take place and a time frame within which visit will be covered by the health insurance (Table 1). Outside the recommended time frame, the check-up can still take place, but must be paid for by the parents [4].

In early 2020, an outbreak of a coronavirus, which was subsequently designated SARS-CoV-2, spread around the globe, becoming known as the COVID-19 pandemic. The infection was transmitted easily and rapidly, and after the first internationally reported case (31 December 2019), it took only 2–3 months to become a pandemic [5]. This caused governments to introduce a variety of measures, among them enforced social distancing, working remotely where possible, and closure of all but the most essential shops (collectively known as “lockdown”). It might also have reduced the opportunity for patients to visit doctors’ practices. While there is no reasonable opportunity to delay essential medical treatment, it is possible and even likely that preventative measures were postponed or even foregone altogether [6]. The World Health Organization (WHO) has stated that routine immunization programs were substantially disrupted in at least 68 countries, affecting more than 80 million children worldwide [7]. The reduction in child vaccination coverage, even for brief periods during emergencies, could lead to an increased number of susceptible individuals and raise the risk of outbreak-prone vaccine-preventable diseases (VPDs) such as measles, polio, and pertussis [8,9,10,11]. Outbreaks of VPDs could be dreadful for a health system that is already battling the impacts of COVID-19, and would substantively increase morbidity and mortality in the age groups most at risk [8].

In Germany, no overall monitoring systems exist that could provide concurrent prescription data for the vaccines (no live monitoring). The aim of this study is to assess the impact of the COVID-19 pandemic on the utilization of preventative paediatric care, focusing on regular check-ups and vaccinations. The present study was conducted to determine the follow-up rate and time between events in paediatric preventative healthcare in Germany before versus during the COVID-19 lockdown. This was accomplished using retrospective data from the IQVIA^TM^ Disease Analyzer database, concentrating on the subpanel of paediatric practices since this speciality covers almost all the events analysed in this study, including paediatric check-ups and vaccinations.

## 2. Materials and Methods

### 2.1. Database

IQVIA^TM^ Disease Analyzer is based on data from German outpatient practices and covers both general practitioners as well as specialists, including paediatricians. The coverage varies by speciality, and at time of extraction of the study data, it was 4.8% for the paediatricians. The database extends back to 1/1992, with the data updated monthly, and contains information on patient demography, diagnoses, prescriptions, laboratory tests, referrals, and sick notes. It is backed up by billing counts, by means of which doctors are reimbursed for their patient treatment. The latter provide more precise information on paediatric check-ups and vaccinations, since diagnosis ICD codes are imprecise or lacking for both, and for vaccinations, there are no prescriptions since they are generally taken from the doctor’s stock [13].

### 2.2. Practice/Patient Selection

Practices were selected if they belonged to the speciality “Paediatric practice” and supplied data continuously for all months of the main analysis periods. Patients were included if they satisfied the following criteria:General criteria: Visiting one of the selected practices, insured by a statutory health insurance, with a U3 check-up (Table 1) at any time in the database history.For baseline count inclusion: With an initial event in the first seven months of one of the analysis periods.For return rate/time between even analyses: satisfying Point 2, and with a follow-up event in the overall analysis period.

### 2.3. Patient Demography

The patients’ ages in the database are anonymised for data privacy reasons; thus, only the year of birth, rather than the precise date, is given. Since this would prove too vague for the purpose of this study, a more precise estimate was obtained from the U3 check-up, which should have taken place around Week 4–5 after birth (maximum range for health insurance reimbursement: Week 3–8). It was thus assumed that the patient was exactly 5 weeks (35 days) old on the date of the U3 check-up. Patient sex is represented precisely in the database. For the hexavalent and pneumococcal vaccinations, the gestation period of the patient was also determined via diagnoses (ICD10 P07.2: severely premature < 27 weeks’ gestation; P07.3: premature < 27–38 weeks’ gestation).

### 2.4. Analysis Timeframes

The main period of analysis was during the COVID-19 pandemic, when restrictions were in place (starting March 2020). At the time of the analysis, data up to September 2021 was available, so the main period lasted from March 2020–September 2021. For comparison, the most recent equally long period before the pandemic was chosen. On account of the fact that January and February 2020 had already been affected by COVID-19, albeit without lockdown, the pre-COVID-19 phase was designated as June 2018–December 2019, both periods therefore spanning 19 months.

For the time between event analyses, it was important that both the initial as well as the follow-up event had taken place in one or the other period, but not in both. For the follow-up event to be observed with sufficient probability within the period being analysed, the initial event was required to have taken place within a seven-month index period at the start of each main period (June 2018–December 2018 or March 2020–September 2020).

### 2.5. Outcomes

The outcomes of interest were the paediatric check-ups (Table 1) carried out by paediatricians in patients up to the age of six, beginning with the check-up visit U3 (4–5 weeks after birth) and followed by the check-up visits U4, U5, U6, U7, and U7a, as well as the main vaccinations taking place in the first six years of the patient’s life. The first two check-ups (U1, U2) are always carried out in hospital and were not eligible for analysis. The vaccinations chosen for analysis were hexavalent (Diphtheria–Pertussis–Poliomyelitis–Tetanus–Hepatitis-B–Haemophilia), pneumococcal, Rotavirus, Quadrivalent (MMR-V: Measles–Mumps–Rubella–Varicella) vaccinations, and these were selected according to the STIKO recommendations for children in Germany (Table 2). Two of the four vaccinations consisted of multiple components and could, in theory, be substituted for by subcomponent vaccinations. The hexavalent vaccination is equivalent to a pentavalent plus hepatitis B vaccination, and the MMR-V to an MMR plus varicella vaccination. For this reason, such combinations were considered to be equivalent and were combined during the analyses to an artificial higher valent vaccination, provided both subcomponents were given on the same day.

For each outcome, initial events in the index periods were identified and counted, and the associated descriptive statistics were generated (patient sex; age at event; for check-ups, concomitant vaccinations of interest on same day; for the Hexavalent and pneumococcal vaccinations, whether patients were born prematurely; and for the Hexavalent and MMR-V vaccinations, of the presence of artificially combined subcomponent vaccines). For all such initial events that were found, subsequent events in the overall period were sought. The subsequent event for the check-ups was always the next one in line (e.g., U4 for U3, U5 for U4, etc.). For vaccinations, the follow-up event was the second or (where indicated) third administration in the vaccination series. It should be noted that the initial event for vaccinations was always the first in series, even if the subsequent one was the third. For these event pairings, the proportion of patients with an initial event who also had a subsequent event (follow-up rate), as well as the statistics representing the time between events and all other previously listed statistics, were determined.

### 2.6. Statistical Methods

For all outcomes, the differences between the pre-COVID-19 and COVID-19 phases were tested statistically. Descriptive statistics are presented here for the outcome variables and covariates, split by group (non-COVID-19 vs. COVID-19). Distributions are shown for sex, gestation period, and vaccination equivalence, and descriptive statistics (mean, standard deviation, minimum, lower quartile, median, upper quartile, and maximum) are shown for time between events and patient age. For distinct distributions (follow-up rate, sex, share of prematurely born patients, and concomitant vaccinations), chi-square tests were carried out, whereas for continuous distributions, the Mann–Whitney U-test was chosen since the assumption of normality was frequently violated.

The statistical software SAS 9.4 (SAS Institute Inc., Cary, NC, USA) was used to select the study cohorts, create the study dataset, and test the descriptive statistics.

Raw patient counts (number of patients with initial or follow-up events) were never tested, since these are dependent on a variety of other factors, most notably the number of patients “at risk”, i.e., those that could theoretically have experienced the event. This may have differed between the two time periods and may have introduced bias; thus, for a meaningful analysis, it would need to have been controlled for. However, the share of patients with follow-up in relation to those with initial events was assessed by means of an odds ratio analysis (logistic regression), since the “at risk” patient counts were known. All other variables were assessed for the statistical significance of the difference between the two time periods (non-COVID-19 vs. COVID-19). Patient sex, duration of gestation period, and vaccine equivalence were analysed by chi-square test, whereas time between events and patient age, which were both continuously distributed, were compared by Student’s t-test or, if the assumption of normality and homogeneity of variance (equal scatter around the mean) were violated, by the nonparametric Mann–Whitney U-test. Due to the fact that the timing of the check-ups was restricted by the health insurance reimbursement policy, there may have been clumping at the extreme values. For example, the anticipated delay due to COVID-19 may have led to an increased number of U3 check-ups close to or at 8 weeks, but higher values than this were not possible. This could have distorted the distribution away from the normal (Gaussian) curve so that the parametric t-test would give unreliable results.

## 3. Results

A total of 165 paediatric practices which continuously supplied data were identified, and these had been visited by 92,295 and 91,746 patients in the pre-COVID-19 and COVID-19 phases, respectively. These patients had a U3 check-up in their history for precise ageing (52,664 and 54,367 patients in the respective index periods). It should be noted that the patients included in these counts could have visited in both periods.

The sex ratio was persistently in favour of males for the paediatric check-ups as well as the vaccinations, but the differences between the pre-COVID-19 and COVID-19 periods always failed to reach significance (Table 3 and Table 4). There was no difference in patient age between the two phases for the U3 check-up, which was by design (all patients were set to 5 weeks = 35 days old at that event). For the remaining check-ups, the differences were minimal, with the greatest difference observed at the U7 check-up (0.79 weeks; pre-COVID-19: 104.40 weeks; COVID-19: 105.19 weeks). For the U4 and U5 check-ups, patients were older in the pre-COVID-19 period, whereas for the latter (U6, U7) check-ups, it was the other way around (Table 3).

For the paediatric check-ups, there was a small to moderate drop in follow-up rates due to the pandemic (Table 3). While the reduction in patients with follow-up events was slight for the U3–U4 comparison (−0.9%), this increased with successive check-ups (U4–U5: −2.7%; U5–U6: −4.2%; U6–U7: −14.3%; U7–U7a: −13.3%).

Although the follow-up rate for paediatric check-ups was reduced by the pandemic, the time between check-ups did not increase. On the contrary, patients returned distinctly earlier for U7 and U7a after their preceding check-ups during COVID-19 (reduction of 1.52 and 1.66 weeks, respectively) and only for the U5–U6 comparison was there a minimal increase (0.13 weeks) in the time between events due to COVID-19.

The proportion of paediatric check-ups with vaccination on the same day was consistently higher during the pandemic. The smallest difference was observed for the U3 (0.5%) and the highest for the U6 (10.0%) check-up, with the others having intermediate differences (U4: 1.7%; U5: 3.3%; U7: 1.6%). Somewhat greater than the differences between phases were the differences between check-ups, with the U4 having the highest (pre-COVID-19: 85.1%; COVID-19: 86.8%) and the U3 the lowest (pre-COVID-19: 3.4%; COVID-19: 3.9%) share of concomitant vaccine administration. The U6 check-up was also associated with a very high degree of concomitant vaccine administration (pre-COVID-19: 70.4%; COVID-19: 80.4%) followed by the U5 (pre-COVID-19: 37.2%; COVID-19: 40.5%) and U7 (pre-COVID-19: 16.3%; COVID-19: 17.9%) check-ups (Table 3).

The age differences between the two phases were greater for vaccinations, exceeding a week for the patients with 1st–3rd Hexavalent (1.21 weeks) and 1st–2nd pneumococcal (1.19 weeks) initial vaccination events. In general, the patients were older when given the initial vaccination pre-COVID-19, but for the 1st–3rd Hexavalent and the MMR-V vaccinations, it was the other way around. There was no difference in age in the patients given initial Rotavirus vaccinations (Table 4).

The follow-up rates differed to a far greater extent for all vaccinations except for Rotavirus, and among these, the return rate was distinctly greater in the COVID-19 phase (Table 4). The increase in follow-up rate during COVID-19 was particularly high for the 1st Hexavalent (+38.3%) and pneumococcal (+37.3%) vaccinations. It is also remarkable that, except for Rotavirus, the lowest follow-up rate in the COVID-19 phase (1st–3rd pneumococcal: 67.1%) was higher than even the highest rate in the pre-COVID-19 phase (MMR-V: 62.4%). The return rate for Rotavirus was high in both phases for the 1st–2nd vaccination (pre-COVID-19: 93.2%; COVID-19: 92.8%), but low for the 1st–3rd vaccination (pre-COVID-19: 39.4%; COVID-19: 37.8%).

With respect to time between vaccinations, the differences between the phases were more marked and inconsistent. Practically no differences were observed for the Rotavirus vaccination (0.08- and 0.35-week reductions in COVID-19, respectively, for 1st–2nd and 1st–3rd). For the Hexavalent vaccination, distinct increases in time between administrations were noted, particularly for 1st–3rd (11.33-week increase in COVID-19), but also for the 1st–2nd injection (1.35-week increase). Distinct decreases in time between events were, however, seen for the 1st–2nd (2.80 weeks) and 1st–3rd pneumococcal (1.45 weeks) vaccinations, as well as the 1st–2nd MMR-V vaccinations (2.35 weeks).

No differences between the phases were noted in the length of the gestation period for the Hexavalent vaccination, and the share of prematurely born babies was close to 4.0% in both phases. This share was similar for the pneumococcal vaccination in the 1st–2nd, but higher for the 1st–3rd vaccination. For the latter comparison, the difference between the pre-COVID-19 (5.5%) and COVID-19 phases (4.5%) was also statistically significant. Regarding the combination of subvalent vaccines for the Hexavalent and MMR-V vaccinations, it was observed that the use of subcomponent products was consistently low (0.3–0.4%) for the first Hexavalent vaccination, and did not differ between the phases. However, for the initial MMR-V vaccination, more than half of the vaccine administrations were made up of combined MMR + Varicella vaccines in both the pre-COVID-19 (50.8%) and COVID-19 phases (59.4%). This also represents a distinct increase in the use of subvalent vaccines during the pandemic (Table 4).

## 4. Discussion

In evaluating the results, certain differences between the pre-COVID-19 and the COVID-19 phases, over and above the direct effects of the lockdown, must be taken into account. These include (i) the extension of the permitted patient age at check-up without loss of health insurance reimbursement for the U6, U7, and U7a check-ups [14]; (ii) the official recommendation for the Hexavalent vaccination to drop the second in the series [3]; and (iii) the reduced availability of MMR-V vaccines [15]. The first was a direct consequence of the pandemic. In May 2020, the governing bodies decided to extend the period within which check-ups could still be publicly reimbursed by 3 months (13 weeks) for the U6 and higher check-ups in order to reduce the pressure on parents and doctors with respect to visits to the practices. This amendment lasted throughout the remainder of the analysed period. The second was introduced in June 2020 and, again, affected the remainder of the COVID-19 period. This was the subject of this study, the main effect being the dramatic lengthening of the expected time between events for the first to third Hexavalent vaccinations. The third difference resulted from the fact that, in Germany, there are two MMR-V products on the market, and one of these was available in significantly reduced quantities from October 2018–July 2020, i.e., for most of the pre-COVID-19 and some of the COVID-19 phase. It would, therefore, be expected that this was compensated to at least some extent by subcomponent vaccines (MMR + V in free combination).

The age at event for the U3 check-up was standardised to 5 weeks so that no variation was expected, but it differed slightly for the other check-ups. This was even the case for the U6, U7, and U7a check-ups; the permitted age was extended by three months during the pandemic without loss of reimbursement. Therefore, it seems as if this legislatory relaxation was not taken advantage of by parents and/or paediatricians. The only sign that parents benefitted from this legislatory change was the fact that patient ages were minimally older pre-COVID-19 for the U4 and U5 check-ups, but for the subsequent ones, the trend was reversed. In contrast, patient age had a far greater distribution in the vaccination data, as shown by the distinctly greater standard deviations. This reflects the fact that paediatric check-ups are only reimbursed within a well-defined range of patient ages, whereas vaccinations may be given at any time.

The reduction in follow-up rates for paediatric check-ups during the pandemic period indicates that some of these follow-ups were not conducted. In most federal states, there is no compulsion for the child to have these, although they are highly encouraged; only Hesse, Baden-Wuerttemberg, and Bavaria [16] have an appropriate legal requirement. If a check-up is omitted, the child can still be brought to the one next in line at the appropriate time, and this will still be reimbursed. This indicates a definite negative effect of the COVID-19 pandemic: a higher share of check-ups were prevented by the lockdown.

The results of the times between check-ups reflect those of the corresponding ages at the event. There was practically no difference in time between events for the U3 and U6 check-ups. In light of the extension of the reimbursable timeframe for the U6, U7, and U7a check-ups, it would be expected that the time between events for the U5–U6, U6–U7, and U7–U7a comparisons had increased during the COVID-19 phase. This was not the case: while the U5-U6 comparison was unaffected, the mean time between events for the other two actually decreased by about 1.5 weeks each during the pandemic. One might think that, if the initial check-up in the comparison were delayed, this would give rise to a reduced period of time between events if the second were conducted at the “normal” time. This, however, would imply that the mean age at initial event would be greater, and this was also not the case here. The logical conclusion is that the exemption was not taken advantage of by parents, and that other factors prompted them to have check-ups conducted earlier rather than later, unless they were not carried out at all (as indicated by the lower follow-up rate).

While the follow-up rate for check-ups was hardly affected by the pandemic, there was a distinctly higher share of patients returning for a second or third vaccination visit in the COVID-19 phase.

One can only speculate on the reasons for this, but it is quite possible that the pandemic increased the population’s awareness of vaccinations in general. Moreover, there is no sufficient evidence of a significant change in parental willingness to routinely vaccinate their children during the COVID-19 pandemic [17]. In addition, parents may well have wished to rule out the possibility of other diseases to the maximum possible extent in case their child became afflicted with SARS-CoV-2. It should be remembered that although the analysis period did extend into the period during which vaccinations against COVID-19 had been developed, it nevertheless mostly comprised the time before these had been made available to the general public. Furthermore, the COVID-19 vaccinations were not indicated at that time for use in children aged under 12 years. It can, therefore, be ruled out that the high follow-up rate for the standard paediatric vaccinations was caused by concurrent vaccination in children mainly vaccinated against SARS-CoV-19.

The time between successive vaccinations for a particular vaccine was generally reduced by the COVID-19 pandemic, but increased for Hexavalent vaccines, and particularly for the time between the first and third vaccinations. This very likely reflects the change in legislation due to which the requirement for the second injection was dropped, except for in prematurely born babies [3], and the time between vaccinations artificially increased for children with normal gestation periods. The share of prematurely born babies observed in this study (ca. 4%) was too low to have an impact, and the relaxation of this ruling counteracted the probable effect of the pandemic that was seen in pneumococcal vaccinations. The fact that what had been the fourth Hexavalent vaccination in the series at 11 months now became the third, thus increasing the expected time between the first and third vaccinations from two (M2–M4) to nine months (M2–M11). The results of the present analysis confirm this in real life.

There was, however, almost no effect on the time between successive Rotavirus vaccinations, with the reduction caused by the pandemic amounting to less than a day (0.08 weeks) between the first and second vaccinations and to slightly more than two days (0.35 weeks) between the first and third vaccinations. This likely reflects the fact that, unlike the other vaccinations, there is an upper limit of eight months for the indicated age at vaccination. This had a similar effect as the restricted reimbursement age for the paediatric check-ups. The Rotavirus vaccination also showed a distinct difference in follow-up rate for the second versus third vaccinations. Although >90% of patients were vaccinated a second time, this dropped to <40% for the third vaccination. This can be explained by the fact that there are two available products on the German market; one of these is indicated for a series of two, while the other is a three-dose schedule.

The proportion of multivalent vaccinations made up of subcomponent vaccines was consistently low for the Hexavalent vaccination, but high for the MMR-V. This reflects the situation in Germany, where there is an official recommendation by the STIKO to administer the initial MMR-V vaccination using subcomponents in different arms to reduce the risk of vaccination-induced febrile seizures [18]. The proportions observed herein, however, are distinctly higher than usually expected, and it also seems puzzling that such a large share of second vaccinations in the pre-COVID-19 phase were given this way. This was probably influenced by the fact that there are two MMR-V vaccinations on the market and that one of these was not available from October 2018–July 2020 [15], i.e., for most of the pre-COVID-19 and some of the COVID-19 analysis period. During this time, doctors either used the other more intensively where possible or relied on subcomponent vaccinations, even when this was not necessarily indicated.

The share of paediatric check-ups with concurrent vaccinations was consistently and significantly higher during the COVID-19 phase than before, and this is another indication that the pandemic increased the general population’s awareness of the benefits of and need for vaccines. However, even more significant than the differences between the phases were the differences between check-ups. The two check-ups mostly associated with concurrent vaccinations were the U4 and U6 check-ups. This is hardly surprising, since the former of these coincides with the first and second Hexavalent and pneumococcal vaccinations as well as all three of the Rotavirus vaccinations, whereas the latter is timed around the first MMR-V vaccination. The co-occurrence of vaccinations with other check-ups again reflects the fact that vaccination series may be started at any time, and the follow-ups occur in relation to the initial vaccination rather than at fixed ages.

Our results regarding the uptake of paediatric vaccination during the COVID-19 pandemic are in line with the vaccination rates for paediatric vaccination in Germany in 2020, which were published by the RKI. They also show no negative impact of the COVID-19 pandemic compared to the vaccination rates in 2019 [19].

The IQVIA^TM^ Disease Analyzer database proved to be an excellent source of information, providing large numbers of subjects with practically no investment of time, since the data was retrospective. When assessing the results, a distinction must be made between statistical significance and clinical relevance. It is questionable whether, for example, the reduction in the time between U3 and U4 check-ups of about a quarter of a week, i.e., less than two days, between the pre-COVID-19 (9.85 weeks) and COVID-19 (9.59 weeks) phases is really clinically relevant, even though it is highly statistically significant (*p* < 0.0001). In view of this, the *p*-values presented herein should be interpreted with a certain degree of caution and regarded as secondary to the parameter differences between the phases.

## 5. Limitations

The main limitation of the IQVIA^TM^ Disease Analyzer database is the fact that patients cannot be observed across practices. If they do appear in two panel practices, it is with two different Patient IDs, so that there will be a duplication as well as a fragmentation of the patient history. While it is possible for patients to switch their standard doctor, and this may have occurred within one of the analysis periods, this would only have an impact on the single events. The analyses of time between events would not be affected by misclassification, only by distortion, since in the case of paediatric check-ups, all are definitely identifiable. For vaccination series, there is no possibility of the third vaccination in a series being mistaken for the second on account of the classification system (Suffix B vs. A to the billing count). The patient would merely be omitted from some of the analyses of the time between events, possibly affecting the representativity of those that are included. This factor is considered to be of minor importance.

One limitation of the methodology rather than the database is that the analysis periods were somewhat short for adequate analysis of the times between successive check-ups. The maximum possible duration is from the earliest possible age of the initial to the latest possible age of the subsequent event. This amounted to nine months for the U5–U6 comparison and less than this for the earlier ones; however, the duration was 18 months for each of the U6–U7 and U7–U7a check-ups. If one takes into account the fact that the maximum age for these was extended by three months during the pandemic, these latter two comparisons were undoubtedly subjected to censoring through the non-identification of very late follow-up events, which lowered the follow-up rates and the mean times between events. It should, however, be remembered that this limitation applies to all of the time periods between vaccinations, since there is no upper limit for these set by the authorities, and the times between events found in this study are restricted more by medical reasons (a severe delay usually results in a repeat of the entire vaccination series).

## 6. Conclusions

The principal conclusion is that the COVID-19 pandemic had little effect on paediatric check-ups, probably because these are rigidly timed and legislated. While many studies from other countries reported disruptions in routine paediatric vaccination schedules, this was not the case in Germany. This study suggests that there was increased engagement with vaccination services, possible indicating that the public awareness of the general need for and benefits to be gained from vaccination increased.

Furthermore, this study shows that retrospective real world databases closely reflect the everyday healthcare situation, and can provide valuable insights into health care beyond the carefully controlled, and, therefore, predetermined, scenarios of most clinical studies.

## Figures and Tables

**Table 1 vaccines-11-00720-t001:** Summary of paediatric check-ups to be conducted in the first six years of a child’s life [12].

Designation	Intended Time Point in Child’s Life	Lower Permissible Limit of Check-Up for SHI Reimbursement	Upper Permissible Limit of Check-Up for SHI Reimbursement
U1	Directly after birth		
U2	3rd–10th day	3rd day	14th day
U3	4th–5th week	3rd week	8th week
U4	3rd–4th month	2nd month	4.5th month
U5	6th–7th month	5th month	8th month
U6	10th–12th month	9th month	14th month
U7	21st–24th month	20th month	27th month
U7a	34th–36th month	33rd month	38th month
U8	46th–48th month	43rd month	50th month
U9	60th–64th month	58th month	66th month

**Table 2 vaccines-11-00720-t002:** Summary of vaccinations recommended for children in the first 18 months of a child’s life and details of the number and timing of doses [3].

Vaccination	Expected Timing of Vaccination after Birth		
	Initiation	2nd in Series	3rd in Series
Hexavalent (Tetanus, Diphtheria, Pertussis, Poliomyelitis, Hib, HepB) *			
up to June 2020	2nd month	3rd month	4th month
after June 2020	2nd month	4th month	11th month
Rotavirus	6th–8th week	3rd month	4th month
Pneumococci	2nd month	4th month	11th month
Tetravalent, MMR-V (Measles, Mumps, Rubella, Varicella)	11th month	15th month	–

* This reflects the most recent STIKO recommendation, whereby patients should be vaccinated at 2, 4, and 11 months (2 + 1 scheme) unless born prematurely (3 + 1 scheme). Up to June 2020, the recommendation was for all patients to be vaccinated according to the 3 + 1 scheme, which envisages an additional injection at 3 months.

**Table 3 vaccines-11-00720-t003:** Results of the analysis of paediatric check-ups for each pair of initial and follow-up events. Percentages of *N* with follow-up are given in relation to *N* with initial event. For other categories, the percentages are in relation to *N* with follow-up.

Parameter	Outcome	Pre-COVID-19	COVID-19	*p*-Value
U3 check-up	*N* with initial	14,490 (100%)	14,299 (100%)	–
	*N* with follow-up	13,152 (90.8%)	12,850 (89.9%)	0.0099
Patient sex	Male	6739 (51.2%)	6534 (50.8%)	0.5282
	Female	6413 (48.8%)	6316 (49.2%)	
Age at U3 (weeks)	Mean (std. dev.)	5.00 (0.00)	5.00 (0.00)	1.0000
	Median (Q1–Q3)	5 (5–5)	5 (5–5)	
Same-day vaccination	With vaccination	443 (3.4%)	507 (3.9%)	0.0131
	No vaccination	12,709 (96.6%)	12,343 (96.1%)	
Time U3–U4 (weeks)	Mean (std. dev.)	9.85 (2.24)	9.59 (2.31)	<0.0001
	Median (Q1–Q3)	10 (9–11)	10 (8–11)	
U4 check-up	N with initial	12,771 (100%)	12,672 (100%)	–
	N with follow-up	11,914 (93.3%)	11,487 (90.6%)	<0.0001
Patient sex	Male	6064 (50.9%)	5795 (50.4%)	0.4915
	Female	5850 (49.1%)	5692 (49.6%)	
Age at U4 (weeks)	Mean (std. dev.)	14.71 (2.25)	14.58 (2.28)	<0.0001
	Median (Q1–Q3)	15 (13–16)	15 (13–16)	
Same-day vaccination	With vaccination	10,144 (85.1%)	9974 (86.8%)	0.0002
	No vaccination	1770 (14.9%)	1513 (13.2%)	
Time U4–U5 (weeks)	Mean (std. dev.)	13.20 (2.85)	12.98 (2.97)	<0.0001
	Median (Q1–Q3)	13 (11–15)	13 (11–15)	
U5 check-up	N with initial	11,885 (100%)	12,304 (100%)	–
	N with follow-up	11,059 (93.1%)	10,934 (88.9%)	<0.0001
Patient sex	Male	5684 (51.4%)	5534 (50.6%)	0.2447
	Female	5375 (48.6%)	5400 (49.4%)	
Age at U5 (weeks)	Mean (std. dev.)	27.93 (2.84)	27.80 (2.87)	0.0007
	Median (Q1–Q3)	28 (26–30)	28 (26–30)	
Same-day vaccination	With vaccination	4118 (37.2%)	4432 (40.5%)	<0.0001
	No vaccination	6941 (62.8%)	6502 (59.5%)	
Time U5–U6 (weeks)	Mean (std. dev.)	23.51 (4.09)	23.64 (4.18)	0.0300
	Median (Q1–Q3)	23 (21–26)	24 (21–26)	
U6 check-up	N with initial	12,166 (100%)	12,441 (100%)	–
	N with follow-up	10,825 (89.0%)	9299 (74.7%)	<0.0001
Patient sex	Male	5596 (51.7%)	4781 (51.4%)	0.6909
	Female	5229 (48.3%)	4518 (48.6%)	
Age at U6 (weeks)	Mean (std. dev.)	51.54 (3.73)	51.69 (3.86)	0.0038
	Median (Q1–Q3)	51 (49–54)	51 (49–54)	
Same-day vaccination	With vaccination	7618 (70.4%)	7481 (80.4%)	<0.0001
	No vaccination	3207 (29.6%)	1818 (19.6%)	
Time U6–U7 (weeks)	Mean (std. dev.)	53.19 (5.79)	51.67 (5.63)	<0.0001
	Median (Q1–Q3)	53 (50–56)	52 (49–54)	
U7 check-up	N with initial	10,739 (100%)	10,970 (100%)	–
	N with follow-up	9333 (86.9%)	8076 (73.6%)	<0.0001
Patient sex	Male	4814 (51.6%)	4147 (51.3%)	0.7613
	Female	4519 (48.4%)	3929 (48.7%)	
Age at U7 (weeks)	Mean (std. dev.)	104.40 (5.38)	105.19 (5.88)	<0.0001
	Median (Q1–Q3)	104 (101–107)	105 (102–108)	
Same-day vaccination	With vaccination	1524 (16.3%)	1446 (17.9%)	0.0058
	No vaccination	7809 (83.7%)	6630 (82.1%)	
Time U7–U7a (weeks)	Mean (std. dev.)	53.15 (6.25)	51.49 (6.16)	<0.0001
	Median (Q1–Q3)	53 (50–56)	52 (48–55)	

**Table 4 vaccines-11-00720-t004:** Results of the analysis of paediatric vaccinations for each pair of initial and follow-up events. Percentages of *N* with follow-up are given in relation to *N* with initial event. For other categories, the percentages are in relation to *N* with follow-up.

Parameter	Outcome	Pre-COVID-19	COVID-19	*p*-Value
Hexavalent (1st–2nd)	*N* with initial	23,394 (100%)	13,068 (100%)	–
	*N* with follow-up	13,349 (57.1%)	12,465 (95.4%)	<0.0001
Patient sex	Male	6816 (51.1%)	6365 (51.1%)	0.9962
	Female	6533 (48.9%)	6100 (48.9%)	
Age at 1st vaccination (weeks) *	Mean (std. dev.)	13.25 (7.63)	12.36 (4.96)	<0.0001
	Median (Q1–Q3)	11 (10–14)	11 (10–14)	
Gestation period	Normal (>37 weeks)	12,808 (95.9%)	11,995 (96.2%)	0.2374
	Short (>28 to ≤37 weeks)	530 (4.0%)	465 (3.7%)	
	Very short (≤28 weeks)	11 (0.1%)	5 (0.0%)	
Vaccine combination	Hexavalent	13,290 (99.6%)	12,411 (99.6%)	0.9151
	Pentavalent + HiB	59 (0.4%)	54 (0.4%)	
Time 1st–2nd injection	Mean (std. dev.)	7.06 (6.59)	8.41 (5.88)	<0.0001
	Median (Q1–Q3)	5 (4–7)	7 (5–10)	
Hexavalent (1st–3rd)	N with initial	23,394 (100%)	13,068 (100%)	–
	N with follow-up	12,457 (53.2%)	10,495 (80.3%)	<0.0001
Patient sex	Male	6351 (51.0%)	5338 (50.9%)	0.8550
	Female	6106 (49.0%)	5157 (49.1%)	
Age at 1st vaccination (weeks) *	Mean (std. dev.)	18.40 (6.35)	19.61 (5.85)	<0.0001
	Median (Q1–Q3)	17 (15–20)	19 (16–22)	
Gestation period	Normal (>37 weeks)	11,952 (95.9%)	10,073 (96.0%)	0.8690
	Short (>28 to ≤37 weeks)	497 (4.0%)	417 (4.0%)	
	Very short (≤28 weeks)	8 (0.1%)	5 (0.0%)	
Vaccine combination	Hexavalent	54 (0.4%)	30 (0.3%)	0.0650
	Pentavalent + HiB	12,403 (99.6%)	10,465 (99.7%)	
Time 1st–3rd injection	Mean (std. dev.)	14.55 (9.36)	25.88 (17.5)	<0.0001
	Median (Q1–Q3)	12 (10–15)	15 (10–43)	
Pneumococci (1st–2nd)	N with initial	24,111 (100%)	13,189 (100%)	–
	N with follow-up	13,622 (56.5%)	12,366 (93.8%)	<0.0001
Patient sex	Male	6966 (51.1%)	6308 (51.0%)	0.8379
	Female	6656 (48.9%)	6058 (49.0%)	
Age at 1st vaccination (weeks) *	Mean (std. dev.)	13.94 (8.07)	12.75 (5.98)	<0.0001
	Median (Q1–Q3)	11 (10–15)	11 (10–14)	
Gestation period	Normal (>37 weeks)	13,080 (96.0%)	11,908 (96.3%)	0.3206
	Short (>28 to ≤37 weeks)	532 (3.9%)	453 (3.7%)	
	Very short (≤28 weeks)	10 (0.1%)	5 (0.0%)	
Time 1st–2nd injection	Mean (std. dev.)	14.63 (10.69)	11.83 (6.79)	<0.0001
	Median (Q1–Q3)	11 (9–15)	10 (9–13)	
Pneumococci (1st–3rd)	N with initial	24,111 (100%)	13,189 (100%)	–
	N with follow-up	8444 (35.0%)	8850 (67.1%)	<0.0001
Patient sex	Male	4273 (50.6%)	4495 (50.8%)	0.8058
	Female	4171 (49.4%)	4355 (49.2%)	
Age at 1st vaccination (weeks) *	Mean (std. dev.)	23.41 (6.2)	22.83 (5.81)	<0.0001
	Median (Q1–Q3)	22 (19–26)	21 (19–26)	
Gestation period	Normal (>37 weeks)	7982 (94.5%)	8456 (95.5%)	0.0073
	Short (>28 to ≤37 weeks)	454 (5.4%)	389 (4.4%)	
	Very short (≤28 weeks)	8 (0.1%)	5 (0.1%)	
Time 1st–3rd injection	Mean (std. dev.)	47.37 (13.23)	45.92 (11.85)	<0.0001
	Median (Q1–Q3)	50 (44–55)	47 (42–53)	
Rotavirus (1st–2nd)	N with initial	11,179 (100%)	11,120 (100%)	–
	N with follow-up	10,423 (93.2%)	10,317 (92.8%)	0.1795
Patient sex	Male	5331 (51.1%)	5253 (50.9%)	0.7398
	Female	5092 (48.9%)	5064 (49.1%)	
Age at 1st vaccination (weeks) *	Mean (std. dev.)	9.93 (2.71)	9.92 (2.73)	0.7746
	Median (Q1–Q3)	10 (8–11)	10 (8–11)	
Time 1st–2nd injection	Mean (std. dev.)	5.81 (2.24)	5.73 (2.19)	0.0075
	Median (Q1–Q3)	5 (4–7)	5 (4–6)	
Rotavirus (1st–3rd)	N with initial	11,179 (100%)	11,120 (100%)	–
	N with follow-up	4403 (39.4%)	4208 (37.8%)	0.0179
Patient sex	Male	2237 (50.8%)	2149 (51.1%)	0.8071
	Female	2166 (49.2%)	2059 (48.9%)	
Age at 1st vaccination (weeks) *	Mean (std. dev.)	15.17 (2.77)	15.08 (2.70)	0.1704
	Median (Q1–Q3)	15 (14–16)	15 (13–16)	
Time 1st–3rd injection	Mean (std. dev.)	11.33 (3.33)	10.98 (3.15)	<0.0001
	Median (Q1–Q3)	10 (9–13)	10 (9–12)	
MMR-V (1st–2nd)	N with initial	14,729 (100%)	11,959 (100%)	–
	N with follow-up	9192 (62.4%)	10,124 (84.7%)	<0.0001
Patient sex	Male	4738 (51.5%)	5178 (51.1%)	0.5795
	Female	4454 (48.5%)	4946 (48.9%)	
Age at 1st vaccination (weeks) *	Mean (std. dev.)	54.13 (11.35)	54.82 (15.00)	0.8406
	Median (Q1–Q3)	52 (49–55)	52 (49–55)	
Vaccine combination	MMR-V	4522 (49.2%)	4114 (40.6%)	<0.0001
	MMR + V	4670 (50.8%)	6010 (59.4%)	
Time 1st–2nd injection	Mean (std. dev.)	22.63 (14.58)	20.28 (12.92)	<0.0001
	Median (Q1–Q3)	18 (13–28)	16 (12–24)	

* “Age at first vaccination” is specific to the vaccination in question. Different ages at first vaccination for, e.g., the two Hexavalent vaccination comparisons are due to the fact that the two values were determined from different subsets of the patient collective.

## Data Availability

Data is unavailable due to privacy or ethical restrictions.

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
