# Peer review of "Effect of the COVID-19 Pandemic on Paediatric Check-Ups and Vaccinations in Germany"

_vaccines, 2023, doi:10.3390/vaccines11040720_

Round 1

Reviewer 1 Report

In the paper entitled “Effect of the COVID-19 Pandemic on Paediatric Check-ups and Vaccinations in Germany”, the authors investigated the effect of COVID-19 in paedaitric vaccinations in different periods during the pandemic in Germany. The paper is good and can be published in this journal after some revisions.

1- First of all, the authors should make the abstract shorter and more clear

2- The authors should elaborate more on the introduction and add more references since I think that the authors forgot to cite many pertinent references on the subject. So, I suggest them to review their search and update their references

3- I’m not aware about the German rules but I’m wondering if the study needs an IRB approval.

Author Response

Dear Reviewer

The Abstract is now more clear and shorter (less than 200 words)

The introduction was revised and contains more background and references

There is no need for an IRB approval. This study was conducted according to Guidelines for Good Pharmacoepidemiology Practices. The utilized databases contain only anomymized data and adress all data protection regulations in Germany. Therefore, no ethics review was needed by an independent ethics committee or institutional review board

Reviewer 2 Report

This article was not a study protocol. This is a full research article.

Author Response

The article was fundamentally revised

Reviewer 3 Report

The manuscript report on an interesting analysis about the impact of pandemic restrictions on vaccine uptake in Germany. The topic is interesting, relevant and certainly useful. 

I feel that the quality of the report needs to be improved, as I will outline in detail below.

Abstract

-       Abstract provided is far too long (500 words) and with too many informations in it. It should be a total of about 200 words maximum. Please refer to author’s information.

Introduction

-       The review of the literature is not so thorough and it should be improved. There are many published articles assessing the pandemic impact on vaccination delays in various European countries. 

-       Lines 44-56: please, explain better the German healthcare system and the pediatric checkups mechanism. You can refer here what is examined at each checkup (lines 94-99).

-       Lines 67-71: The aim of study should be rephased and cleared. 

Materials and Methods

-       The title section should be “Materials and Methods”.

-       Some formatting issues (lines 88-92).

-       Methodology section is exhaustive, listing all the vaccinations although seems redundant.

-       Lines 156-160: The description of statistical analyses performed is overly superficial and should be improved.

Results

-       Results section is not linear, it would be better structured if it would explain the two tables one by one, rather than jumping from table to table. 

-       The significant value (P-value) is never reported in the results section. 

-       Even if it is not too manifest, it seems that there is already an opinion about the results, pay attention to the way you use adjectives and strengthen the terms.

-       Even if not too evident, it seems there’s already an opinion about the results, be careful how you use adjectives and strengthening terms.

Discussion and conclusions

-       Most of the reports worldwide reported a decline or delay in vaccination at the time of the COVID-19 pandemic. Please explain better the public health implication related to study data. The immunizations coverage in 2020 in Germany was higher, lower or the same of 2019?

-       Please be mindful of the relationship between vaccine hesitancy and awareness of the importance of vaccines. It represents one of the aspects of vaccine hesitation and delay and others must be considered in the conclusions.

English seems correct enough, but if you chose “paediatric” it means that you must use consistently British English instead of American English, there’s some typos, for examples the use of “pediatric” instead of “paediatric”.

The manuscript requires revision as mentioned above.

Author Response

Dear Reviewer

The Abstract is now more clear and shorter (less than 200 words)

We added more references about the situation of the vaccinations during the pandemic.

The German heathcare system is now better explained

The aim of the study was rephased

The section Material was renamed

Why the P-value is not reported: The main point that speaks against the inclusion of the p-values in the text is that they give the wrong impression in some places.  A reduction of about two days (0.26 weeks) in the period between U3 and U4 (9,.85 weeks vs. 9.59 weeks before/during Corona) which is highly significant (p < 0.0001) shows at best a statistical significance but hardly a clinical relevance. The p-values are not concealed but can be seen in the tables.

The comparison of our data with the VCR's in Germany in 2020 and 2019 was added to the discussion.

Reviewer 4 Report

After carefully reading the paper, some major  comments should

be discussed before publishing as follows:

1. Motivation Section should be added.

2. The physical interpretation should be discussed for the new

scheme.

3. Quotation marks should be followed, especially, for

equations.

4. In keywords Section, the authors used capital “upper” and

small “lower” letters from beginning of words. Check.

5. Abbreviation Section should be listed.

6. Why the proposed scheme is flexible as the authors said?.

Discuss.

7. Non-parametric plots should be sketched to discuss the

data behavior.

8. The conclusion should be re-written to be more fitted

9. what this Median (Q1 – Q3)?

10. Data or a clear source that allows verification of results must be attached

11. Which program is used to obtain these values and results?

12. What is the usual test method? Ie what does that mean P-value smaller than 0.05

13. why used Chi-square tests ?

14. Is it confirmed that the data follows a normal distribution?

15.Attach the raw data file

16. The method used and its purpose must be clarified

Author Response

Dear Reviewer

The motivation was added to the article.

The keyword section was revised

Abbreviation section was added

Non-parametric plots should be sketched to discuss the data behavior:  Nonparametric plots would consume a lot of additional space and only serve to more visually present the information already given in the tables (mean, standard deviation, median, lower and upper quartiles)

The conclusion was revised

What this Median (Q1 – Q3)?  The respective results stand for the median (50th percentile), lower quartile (25th percentile) and upper quartile (75th percentile).  The mode of presentation, though not universal, is pretty standard

Data or a clear source that allows verification of results must be attached: The extracted raw data is contained in large files – it does, after all, pertain to several tens of thousands of patients – and is far too voluminous to be attached in printed form.  Furthermore, this may well infringe on German data privacy legislation so that the possibility of open access would have to be investigated first

Which program is used to obtain these values and results:  The data was extracted using the Vaccine Analyzer interface and imported in the SAS Vs. 9.4 where all analyses were conducted (was added to the section 2.5)

What is the usual test method? Ie what does that mean P-value smaller than 0.05:  There is no “normal” test method – each test method has its own assumptions and the choice of test method must be tailored to the nature of the data (distribution of data, categorical or continuous, etc.).  A p-value <0.05 (=5%) denotes statistical significance.  The limit for α (alpha), here set to 5%, can be lowered to make the test more stringent; however, 5% is the most commonly used limit value

Why used Chi-square tests ?  This was done in order to remain free from the constraint of having to ensure that the data is normally distributed (or, alternatively, adequately normalising the data by an appropriate transformation). See also the following point

Is it confirmed that the data follows a normal distribution? It was at no point confirmed, that the data followed a normal (Gaussian) distribution.  This is the reason why we chose nonparametric statistical tests to make the comparisons since these do not rely on this assumption

Unfortunately we are not able to attch the raw data file. Data sets are too large and the release is anyway dependent on the legislation on data protection.

Reviewer 5 Report

The study is well planned and methodologically well developed and contextualized in Germany.

The Abstract is too long and should be shortened. It is preferable not to include acronyms in the abstract.

The keywords used should be reviewed and adapted as much as possible to the MeSH (Medical Subject Headings)

In the introduction section, the information on the German context should be expanded to facilitate the reading of the work by readers from other countries. For example, information on the age covered by pediatric care should be provided, recommended vaccinations should also be included in this section and not in the methodology section.

In the Discussion section the paper should be contrasted with other studies on aspects related to the global impact of COVID-19 on health care during the pandemic.

In my opinion, the conclusions section is poor and is a reiteration of the results obtained.

The references must be corrected and exposed properly.

Author Response

Dear Reviewer

The abstract contains now less than 200 words

The keywords were revised

The introduction section was revised and contains now the recommended informations.

The Discussion section was revised

The conclusions section was re-written

The references were corrected

Round 2

Reviewer 1 Report

The authors made all the required changes. 

Author Response

Dear Reviewer

Thank's for your feedback.

Reviewer 3 Report

The manuscript seems improved in this revision. However, some points can be improved or are been not not take in account by authors:

1. The review of the literature remain not so thorough and it should be improved. These article should be cited:

-Parental Willingness and Associated Factors of Pediatric Vaccination in the Era of COVID-19 Pandemic: A Systematic Review and Meta-Analysis. Wang Z, Chen S, Fang Y 2022 PMID: 36146530 PMCID: PMC9506252 DOI: 10.3390/vaccines10091453

- Parental Hesitancy towards the Established Childhood Vaccination Programmes in the COVID-19 Era: Assessing the Drivers of a Challenging Public Health Concern Derdemezis C, Markozannes G, Rontogianni MO, Trigki M, Kanellopoulou A, Papamichail D, Aretouli E, Ntzani E, Tsilidis KK 2022 PMID: 35632570 PMCID: PMC9144671 DOI: 10.3390/vaccines10050814

- The Effect of the COVID-19 Pandemic on Childhood Immunizations: Ways to Strengthen Routine Vaccination McNally VV, Bernstein HH 2020 PMID: 33290569 DOI: 10.3928/19382359-20201115-01

2. in statistical methods, please rename the subtitle in "2.6 Statistical Methods" (line 151)

3. line 174: consider to substitute chi symbol with "chi-square test"

4. Results section was not improved. The authors should explain the two tables one by one, rather than jumping from table to table. To avoid to give an opinion in results section, the authors should pay attention to the way you use adjectives and strengthen the terms. ( e.g. "slightly" or "minimally" or "little greater" or "practically no difference" or "slight increase" are not appropriate in scientific context)

5. the motivation about how the p-value is presented in the article or table should be explicated in manuscript (discussion section) with author's interpretation as reported in "author response".

Author Response

Dear Reviewer

  1. The articles from Wang et al and McNally et al were included. The articel from Derdemezis does not fit to our publication. The results are in conflict to the results from Wang et al. Furthermore the article shows no comparison regarding the vaccine hestinacy bevor and during the COVID-19 pandemic. I hope for your understanding.
  2. The section "statistical methods" was renamed (line 154)
  3. The chi symbol was substituted (line 177)
  4. The result section was revised. Now we explain the result of the two tables one by one and we revised the used adjectives
  5. The explenation of how the p-values are presented is included in lines 385 - 391.

Reviewer 4 Report

The paper is good now

Author Response

Dear Reviewer

Thank's for your feedback

Reviewer 5 Report

The paper has been substantially improved according to the suggestions.

In relation to the citations included in the references section, I leave it to the discretion of the editorial committee if it considers them correct.

Author Response

Dear Reviewer

Thank's for your feedback